# UniCTokens-R1: Boosting Unified Personalization via Reinforcement Learning

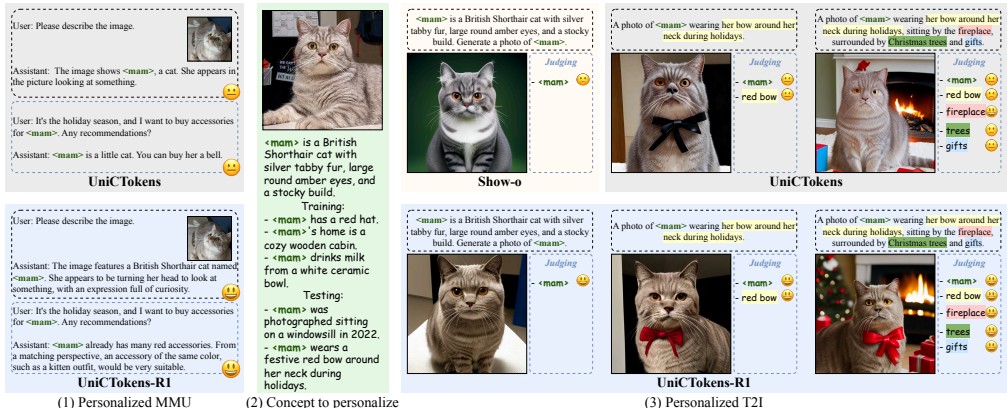

Figure 1: **The capability overview of UniCTokens-R1.** The UniCTokens-R1 method optimizes understanding, reasoning, and generation reasoning using the UniCTask-GRPO method at the same time. By leveraging the mutual promotion between understanding and generation, it successfully achieves a high-level injection of personalized concept information. UniCTokens-R1 can not only perform various personalized reasoning tasks but also excels in dense text generation tasks and reasoning-based generation tasks with missing concept information.

## ABSTRACT

The rapid development of Unified Models demonstrates their potential for personalized understanding and generation tasks. However, existing methods either focus on single tasks or rely on complex training processes to achieve cross-task information sharing, which hinders the model's ability to fully capture user information and its broader real-world applications. In this work, we propose UniCTokens-R1, an end-to-end reinforcement learning framework that facilitates mutual enhancement of understanding and generation. Specifically, the model performs both tasks in a single stage, leveraging the detailed semantic information obtained from the understanding task to assist in generation, and subsequently using the generated results as feedback to improve understanding capabilities. We adopt an optimization method, UniCTask-GRPO, that integrates ensembled rewards to seamlessly optimize both tasks simultaneously. We also propose a novel training strategy that dynamically adjusts the number of generated samples to accelerate convergence. To better model real-world user requests, we expanded the existing UnifyBench from two perspectives: denser descriptions and additional user extra information. Experiments demonstrate that our UniCTokens-R1 achieves state-of-the-art results on UnifyBench++, showcasing model's cross-task information reasoning capabilities.

## 1 INTRODUCTION

With the development of Unified Vision-Language Models (ULM) (Jiang et al., 2025; Team, 2025; Xie et al., 2025a; Li et al., 2025; Xie et al., 2025b; Deng et al., 2025; Wu et al., 2024b), it has achieved impressive performance in general understanding and generation scenarios. These models demonstrate strong ability by handling both understanding and generation tasks within a single

model However, the translation of general-purpose technology into practical applications derives its core value from user-centric personalization. Nevertheless, the current research focus on general capabilities, overlooking the huge potential of models in personalization domain.

User requests in real-world scenarios are rarely isolated, instead, they typically entail a complex interplay between understanding user-provided information and generating appropriately personalized content. In recent years, several studies have begun exploring the application of ULM in personalized tasks (Ma et al., 2025; Nguyen et al., 2025; An et al., 2025b). However, these efforts either overlooks the connections and focus solely on a single task (Jiang & Chen, 2025; Tian et al., 2025), or demand extremely complex methods and incur high costs for model personalization. Together, this highlights two core challenges that unified models face when executing personalized tasks:

• How to better utilize the information to achieve mutual enhancement among personalized tasks.

• For more practical applications, how to obtain personalized model as simple as possible.

To better address these challenges, we propose **UniCTokens-R1**, an end-to-end reinforcement learning framework that facilitates the mutual enhancement of understanding and generation reasoning. Different from previous methods that rely on implicit shared tokens for cross-task information sharing (An et al., 2025b), we attempt to directly bridge the information gap between understanding and generation through explicit reasoning information transfer (Penha et al., 2025). Specifically, we let the model perform personalized understanding and generation tasks simultaneously and optimize their reasoning together. The fine-grained user-provided concept information from answering personalized understanding tasks can serve as a supplement to the generation task, promoting the generation of images that are more consistent with given information. Similarly, the generated images are directly fed to the reward ensemble as the result of the entire process to obtain a reward, which in turn indirectly promotes more comprehensive extraction of provided information.

Although it has been recognized that these two tasks can explicitly promote each other through reasoning, a critical question remains unaddressed: How can we better coordinate these two reasoning tasks for further boosting model personalization capabilities? To meet our needs, we propose **UniCTask-GRPO**, an RL method to jointly optimize the two tasks for ULM. We choose RL instead of supervised fine-tuning (SFT) for two reasons:

• Firstly, the ULM we use already has the basic capabilities of personalized understanding and generation. Therefore, our goal is not to teach the model these skills from scratch, but to elicit and coordinate these two existing capabilities of the model (Chen et al., 2025; Liang et al., 2025). RL is naturally suitable for this goal because it guides the model's self-exploration to find the best strategy for integrating the two capabilities, rather than simply imitating a static dataset like SFT.

• Secondly, RL has proven to be very effective in enhancing complex reasoning abilities (Bai et al., 2022; Hu et al., 2025; Sarch et al., 2025), and strong reasoning ability is the key to achieving information transfer and mutual promotion between tasks. By constructing an incentive mechanism that rewards the synergy between "understanding" and "generation", we can effectively guide the model to produce more logically coherent and sophisticated personalized results.

Since the generation task does not have clear reward rules like the understanding task, we utilize an integrated reward system composed of multiple visual expert models. Besides, to further improve efficiency and better meet the needs of real-world personalized tasks, we also propose a novel method DSOG (Dynamic Scaling of Groupsize), a dynamically adaptive method which ensures the quality of the samples finally used for reward calculation and accelerates the convergence of training.

To better model the complex queries of real-world users, we extended the existing UnifyBench to **Unifybench++**. We challenged the model's ability from two aspects: denser text descriptions and more additional user information. When we tested our UniCTokens-R1, we consistently achieved competitive or better results compared to all other personalization methods. Interestingly, we found that with our framework, the model can achieve the effect of coordinated mutual enhancement between tasks without the need for a complicated cold-start process. Both quantitative and qualitative results indicate that our method enables the model to generate more user-aligned results through cross-task information transfer and shows enhanced robustness in handling personalization scenarios.

Finally, our contributions are summarized as follows:

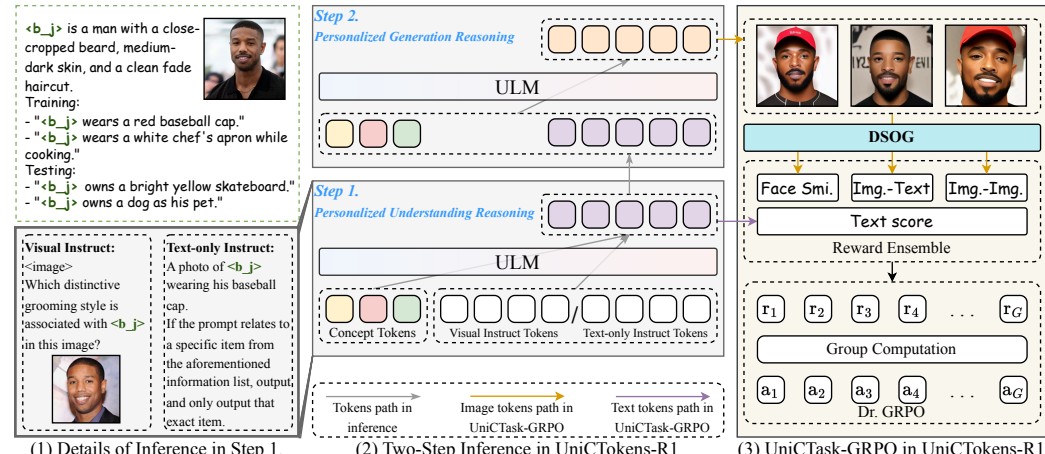

(1) Details of Inference in Step 1.    (2) Two-Step Inference in UniCTokens-R1    (3) UniCTask-GRPO in UniCTokens-R1

Figure 2: **The overview of UniCTokens-R1.** We innovatively integrate personalized understanding reasoning and personalized generation reasoning. Through UniCTask-GRPO, we enable these two processes to mutually promote each other, jointly contribute to parameter updates, enhance the model's reasoning ability regarding the concepts, and achieve a high-level injection of information.

- We propose UniCTokens-R1, an end-to-end RL framework that facilitates the mutual enhancement between personalized understanding and generation.
- We develop UniCTask-GRPO and DSOG to effectively and efficiently coordinate the optimization of both tasks, further boosting model's personalized capability.
- We expands the existing UnifyBench to obtain harder and more robust evaluation. Our experiments demonstrate superior performance over existing methods on all personalization tasks.

## 2 RELATED WORK

**Reinforcement Learning for Generation.** Quantities of works have proved that reinforcement learning (RL) has emerged as one of the core pathways to enhance model practicality and task adaptability (Yuan et al., 2025; Yue et al., 2025). Recently, research has begun exploring the application of RL to generation tasks. Early studies such as Chern et al. (2025) and Pan et al. (2025a) focused on the use of chain-of-thought (CoT) in generation tasks. With the introduction of GRPO (DeepSeek-AI et al., 2025), RL methods have been increasingly adopted in this area(Tong et al., 2025; Han et al., 2025; Xue et al., 2025; Xiao et al., 2025). In multimodal research, RL has also been employed to optimize the synergy between multimodal understanding and generation capabilities of models (Mao et al., 2025; Zhang et al., 2025b). We pioneers the application of such RL-based methods to the personalized domain: we integrate two aspects of personalized reasoning, design an end-to-end RL paradigm UniCTask-GRPO.

## 3 METHOD

To further achieve the mutual promotion of understanding and generation, we propose an end-to-end GRPO reinforcement learning framework, UniCTokens-R1 (as shown in Figure 2), which successfully improves the model's performance on the two tasks and enhances the model's personalized reasoning ability. In this section, we first introduce the definition of personalized reasoning for understanding and generation tasks, then propose the end-to-end UniCTask-GRPO, and finally propose the DSOG method to accelerate the training and introduce the composition of rewards.

### 3.1 PERSONALIZED UNDERSTANDING AND GENERATION REASONING

Existing works such as YoChameleon (Nguyen et al., 2025) and UniCTokens (An et al., 2025b) achieve information sharing between understanding and generation tasks at the token level through

complex training, but this inevitably leads to information loss. In this work, we hope to integrate cross-task information at task level to enhance the personalized reasoning ability of the ULM. Therefore, we propose personalized reasoning in two aspects (shown at (2) in Figure 2) to enable the ULM to have a more comprehensive and detailed grasp of personalized knowledge.

**Personalized Understanding Reasoning.** Personalized understanding reasoning consists of two parts: visual instruct reasoning and text-only reasoning (shown at (1) in Figure 2). For the former, we utilize the $(Image, Qustion, Answer)$ triples in Unifybench++. By inputting the image and the question, we enable the ULM to perform personalized image understanding, thus learning the characteristics of personalized concepts.

$$Image + Question \xrightarrow{\text{ULM}} IR \tag{1}$$

For the latter, we use the expanded part in Unifybench++, several additional pieces of attribute information and corresponding reasoning prompts for each personalized concept (for detailed information, see the Appendix C). When conducting text-only reasoning, the input consists of all the extra information and a certain reasoning prompt $IP$. The ULM needs to understand, and reason about the information, and output the attribute information that best conforms to the reasoning prompt.

$$Extra\_info + IP \xrightarrow{\text{ULM}} IR \tag{2}$$

**Personalized Generation Reasoning.** During the process of personalized understanding reasoning, we obtain the reasoning result of the ULM, denoted as $IR$. For visual instruct reasoning, we concatenate the basic prompt $BP$ with the reasoning result $IR$ to get the final compound prompt $CP$ where $CP = BP + IR$.

$$CP = \text{a photo of } \langle\text{sks}\rangle. + \langle\text{sks}\rangle \text{ is (attributes)} \tag{3}$$

We use $\langle\text{sks}\rangle$ to represent a learnable unique identifier for this new concept. So we can formalize $CP$ as Equation 3, in which $IR$ will contain more detailed descriptions of picture details (for example, $\langle\text{b\_j}\rangle$ has brown eyes), which helps to generate a more realistic image, embodying the concept that understanding promotes generation. For text-only reasoning, we concatenate the reasoning prompt $IP$ and the reasoning result $IR$ to construct the compound prompt, i.e., $CP = IP + IR$. As Equation 4 shows, The ULM will first infer the unclear information in the original prompt $IP$ based on the known knowledge, and then generate the final image. Through subsequent end-to-end UniCTask-GRPO training, we achieve the mutual promotion of understanding and generation, enhancing the model's personalized reasoning ability while improving the capability of personalization. For more details about the two-step reasoning, please refer to the Appendix D.

$$CP = \text{a photo of } \langle\text{sks}\rangle \text{ (Inference Prompt)}. + \langle\text{sks}\rangle \text{ is (Reasoning Results)} \tag{4}$$

### 3.2 UniCTask-GRPO

As an improved algorithm of PPO, GRPO effectively addresses the adaptability issue of PPO in large-model reasoning tasks by removing the dependence on traditional value functions and estimating the advantage value from a "group-relative perspective"(DeepSeek-AI et al., 2025). Many works have shown that the GRPO method has a significant effect on improving the complex reasoning performance of large models (Zhi et al., 2025; He et al., 2025; Ding et al., 2025).However, Liu et al. (2025c) points out that GRPO will cause the model to favor generating longer responses and assign different weights to problems of varying difficulty. Since we have two forms of personalized understanding reasoning, which means there do exist distinctive difficulty. Also, it is harmful to give long prompt in T2I task, which may cause confusion to the model. So we adopt Dr.GRPO method raised by Liu et al. (2025c).The details of Dr.GRPO are shown in the Appendix H.

In this paper, based on the preliminary concept information injection of UniCTokens, we integrate the reasoning in terms of personalized understanding and personalized generation at the task level, and design UniCTask-GRPO, an end-to-end GRPO training method. First, through random sampling, we determine the form of personalized understanding reasoning. Subsequently, we conduct the reasoning process to obtain $IR$. According to the original prompt $BP/IP$, we construct the compound prompt $CP$. Finally, we generate images based on $CP$. Show-o (Xie et al., 2025a) leverages the decoding

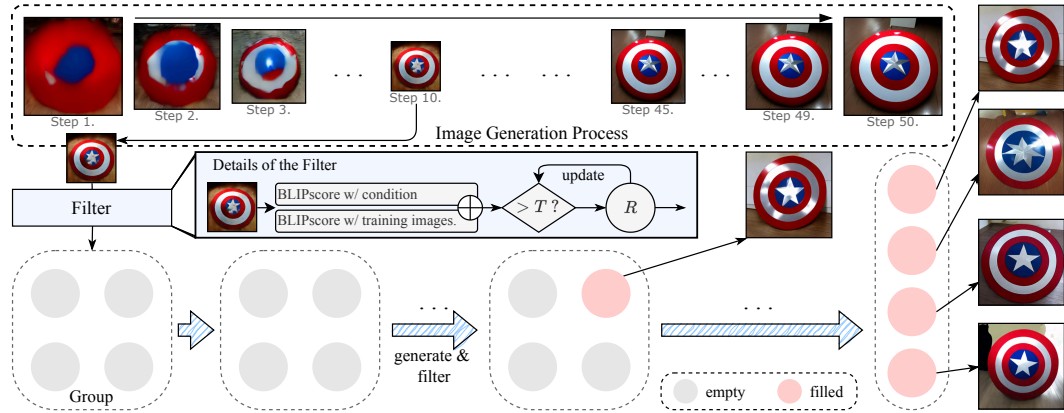

Figure 3: **Illustration of DSOG.** We use the generation result at the 10th step of Show-o to preliminarily judge the effect of the final generated result. Only when the score is higher than the threshold will the generation of the remaining steps be continued. Furthermore, we dynamically update the threshold so that the probability of the score being higher than the threshold remains at a certain level.

methodology from MaskGIT (Chang et al., 2022), obtaining the final image by eliminating masked tokens. For generating an image $I_T$, the process is formulated as:

$$p_\theta(I_0 \mid I_T) = \prod_{t=1}^{T} p_\gamma(I_{t-1} \mid I_t) \tag{5}$$

where $I_t$ denotes the latent variables at step $t$, and $p_\theta(I_{t-1} \mid I_t)$ represents the conditional probability distribution specified by the model parameters $\gamma$.

Therefore, at each timestep, the determined tokens serve as context to influence the prediction of the remaining masked tokens, and all tokens together form a complete description of the current generation state. As a reinforcement learning method, Dr.GRPO requires a complete policy trajectory. Thus, we use the probabilities of all image tokens at each timestep for calculation.

Since one parameter update involves two inferences (denoted as $o_i = (IR_i, I_{i,0}, I_{i,1}...I_{i,T})$), the $D_{i,j}(\theta)$ in the optimization goal in (Appendix H) is converted to:

$$D_{i,j}(\theta) = \frac{\pi_\theta(o_{i,j} \mid q, o_{i,<j})}{\pi_{\theta_{old}}(o_{i,j} \mid q, o_{i,<j})} = \begin{cases} \frac{\pi_\theta(IR_{i,j}|q,IR_{i,<j})}{\pi_{\theta_{old}}(IR_{i,j}|q,IR_{i,<j})}, & 0 \le j \le |IR_i| \\ \frac{\pi_\theta(I_{i,t,j}|CP,I_{<t})}{\pi_{\theta_{old}}(I_{i,t,j}|CP,I_{<t})}, & |IR_i| < j \le |IR_i| + (T+1)|I_{i,T}| \end{cases} \tag{6}$$

Finally, we assign weights to the understanding task and the generation task, and ultimately obtain the optimization objective function formula of UniCTask-GRPO:

$$\mathcal{J}_{\text{UniCTask-GRPO}}(\theta) = \mathbb{E}_{(q,a)\sim\mathcal{D},\{o_i\}_{i=1}^G \sim \pi_{\theta_{old}}(\cdot|q)}$$

$$\left[ \frac{1}{G} \sum_{i=1}^{G} \sum_{j=1}^{|o_i|} \alpha \left( \min \left( D_{i,j}(\theta)\hat{A}_i, \text{clip} \left( D_{i,j}(\theta), 1-\varepsilon, 1+\varepsilon \right) \hat{A}_i \right) - \beta D_{\text{KL}}(\pi_\theta \parallel \pi_{\text{ref}}) \right) \right] \tag{7}$$

where

$$\alpha = \begin{cases} \alpha_{Und.}, & 0 \le j \le |IR_i| \\ \alpha_{Gen.}, & |IR_i| < j \le |IR_i| + (T+1)|I_{i,T}| \end{cases} \tag{8}$$

### 3.3 TRAINING STRATEGY

#### 3.3.1 DYNAMIC SCALING OF GROUP SIZE FOR FASTER CONVERGENCE

Many results have shown that group size is a key parameter in GRPO. However, a larger group size implies longer training time and higher requirements for hardware conditions. In real-world scenarios, we always expect the ULM to master all the personalized knowledge within a short training time.

Shrivastava et al. (2025) proposed that the number of generated samples can be increased and dynamic screening can be carried out among them to improve the quality of the samples participating in the calculation. Inspired by this, we propose DSOG (dynamic scaling of group size) method. As shown in the Figure 3, we can roughly determine the effect of the final generated image by obtaining the temporary image generated in one of the intermediate cycles, so that the final images are all of high-quality, accelerating the convergence of the model.

Through An et al. (2025a), when $t = \frac{1}{5} timestep = 10 \, (timestep = 50)$, the image already takes shape ( Figure 3). So we obtain that image and use the blip evaluation reward (BER) to calculate the score. Only when the score is greater than the threshold (denoted as $T$) can it enter the subsequent UniCTask-GRPO process. By this method, we can increase the group size in a disguised form with a small amount of extra time overhead but achieve an excellent acceleration effect.

$$\Delta_i = \text{TPR} - \text{PR}_i, \quad T_i = T_{i-1}^{1+\mu\Delta_i - \varepsilon\tau_{i-1}\cdot\Delta_i}, \quad \tau_i = \eta\tau_{i-1} + (1-\eta)(T_i - T_{i-1}), \ \eta \in (0,1) \quad (9)$$

$$\varepsilon = \begin{cases} \varepsilon_0, & \tau_{i-1} \cdot \Delta_i > 0 \\ 0, & \text{else} \end{cases} \quad (10)$$

Since the model parameters are gradually updated, we hope that the threshold is adaptable correspondingly. Therefore, we update the threshold value dynamically in the training process. Specifically, as shown in Equation 9-Equation 10, we use $\tau$ to record the changing trend of the threshold, and use $\Delta$ to reflect the gap between the pass rate in the current batch $PR_i$ and the target pass rate $TPR$. The new threshold is affected by $\tau$ and $\Delta$. When the pass rate is small, the threshold will be appropriately decreased, and vice versa. When the change of the threshold violates $\tau$, the influence of $\Delta$ on the threshold will be appropriately obstructed to further enhance robustness and prevent the threshold from changing too quickly. The values of other parameters are shown in Appendix F

### 3.3.2 REWARD ENSEMBLE

Due to the differences of the tasks, we cannot adopt the same reward function as DeepSeek. Instead, we need to comprehensively consider the characteristics of text and images for the design. Finally, We have designed a comprehensive and effective reward function shown as Equation 12. The following is a brief introduction to these rewards. For more detailed introduction, please refer to the Appendix E.

$$Reward = w_1 TIER + w_2 BER + w_3 DER + w_4 FER \quad (11)$$

**Text Inference Evaluation Reward (TIER)** We use ERNIE 3.0 (Sun et al., 2021) to compute the embeddings of the ULM output $IR$ and all concept's extra information. Subsequently, the final score is obtained based on the cosine similarity.

**BLIP Evaluation Reward (BER)** To evaluate whether the generated image correctly reflects the content involved in the text prompt, we use the BLIP-2 model (Li et al., 2023) to calculate the cross-modal similarity between the generated image and the text prompt to obtain the BER score.

**DINOv2 Evaluation Reward (DER)** The DINOv2 model is trained in a self-supervised manner and can accurately extract the main body information from pictures of different scenes and styles(Oquab et al., 2023). Therefore, to evaluate the generation effect of the personalized concept main body, we adopt the DINOv2 model as one of the scoring experts. We calculate the patch-level visual features of the generated image and the reference image, and obtain the final score through cosine similarity.

**Facenet Evaluation Reward (FER)** To improve the human-perceived similarity of the face in the generated image, we adopt the Facenet (Wang & Deng, 2021; Schroff et al., 2015): Based on MCTNN (Zhang et al., 2016), we first extract the key features of the face and transform them to the standard position to solve the errors caused by factors such as the picture angle. Subsequently, we calculate the embedding of the face image and obtain the final reward through cosine similarity.

## 4 EXPERIMENT

In this section, we first introduce the experimental setup in detail in the Section 4.1. In 4.2, we present the main results of UniCTokens-R1 in UnifyBench. Then, in 4.3, we showcase the design and corresponding results of multiple ablation experiments. For more comparison results on understanding- and generation-only benchmarks, as well as more visualizations, please refer to the Appendix F.

| Type | Method | Model Size | Und. | | | | | | | Gen. | | | | | | | |
|---|---|---|---|---|---|---|---|---|---|---|---|---|---|---|---|---|---|
| | | | Rec. | Rea.* | Dense Rea.* | VQA | | QA | | Pure Gen. | | Dense Gen.* | | Rea. Gen.* | | Dense Rea. Gen.* | |
| | | | Weight | BLEU | GPT | BLEU | GPT | BLEU | GPT | CLIP-T | CLIP-I | GPT | CLIP-I | CLIP-T | CLIP-I | GPT | CLIP-I |
| **Upper Bound** | GPT-4o+TP | 200B | 0.742 | 0.746 | 0.819 | 0.784 | 0.863 | 0.611 | 0.699 | 0.306 | 0.684 | 0.374 | 0.697 | 0.357 | 0.776 | 0.403 | 0.692 |
| | GPT-4o+IP | 200B | 0.788 | 0.745 | 0.854 | 0.751 | 0.784 | 0.594 | 0.658 | 0.309 | 0.787 | 0.382 | 0.672 | 0.376 | 0.804 | 0.381 | 0.796 |
| | Real Images | - | - | - | - | - | - | - | - | - | 0.833 | - | - | - | - | - | - |
| **Und. Only** | Yo'LLaVA | 13B | 0.921 | 0.327 | 0.529 | 0.616 | 0.625 | 0.614 | 0.594 | - | - | - | - | - | - | - | - |
| | MC-LLaVA | 13B | 0.924 | 0.297 | 0.511 | 0.623 | 0.636 | 0.606 | 0.583 | - | - | - | - | - | - | - | - |
| | RAP-MLLM | 13B | 0.936 | 0.332 | 0.595 | 0.624 | 0.617 | 0.712 | 0.723 | - | - | - | - | - | - | - | - |
| | Qwen2.5-VL + TP | 3B | 0.669 | 0.218 | 0.341 | 0.404 | 0.726 | 0.579 | 0.770 | - | - | - | - | - | - | - | - |
| | Yo'LLaVA(Phi-1.5) | 1.3B | 0.769 | 0.225 | 0.484 | 0.493 | 0.493 | 0.511 | 0.498 | - | - | - | - | - | - | - | - |
| **Gen. Only** | Text inversion | 1.0B | - | - | - | - | - | - | - | 0.248 | 0.632 | 0.322 | 0.538 | 0.294 | 0.673 | 0.351 | 0.633 |
| | DreamBooth (SD) | 1.0B | - | - | - | - | - | - | - | 0.282 | 0.645 | 0.323 | 0.558 | 0.313 | 0.661 | 0.367 | 0.651 |
| **Unified Model** | Chamaleon+TP | 7B | 0.685 | 0.206 | 0.313 | 0.411 | 0.489 | 0.509 | 0.560 | 0.186 | 0.542 | 0.281 | 0.463 | 0.193 | 0.516 | 0.309 | 0.549 |
| | Chamaleon+IP | 7B | 0.497 | 0.216 | 0.374 | 0.446 | 0.497 | 0.411 | 0.532 | 0.165 | 0.514 | 0.266 | 0.449 | 0.190 | 0.532 | 0.299 | 0.503 |
| | Show-o+TP | 1.3B | 0.562 | 0.217 | 0.337 | 0.462 | 0.412 | 0.507 | 0.579 | 0.263 | 0.663 | 0.299 | 0.558 | 0.235 | 0.679 | 0.341 | 0.660 |
| | Yo'Chameleon | 7B | 0.764 | 0.231 | 0.399 | 0.470 | 0.511 | 0.506 | 0.582 | 0.235 | 0.697 | 0.289 | 0.601 | 0.273 | 0.704 | 0.321 | 0.702 |
| | UniCTokens | 1.3B | 0.792 | 0.238 | 0.385 | 0.505 | 0.521 | 0.546 | 0.601 | 0.280 | 0.750 | 0.298 | 0.639 | 0.282 | 0.762 | 0.317 | 0.712 |
| | UniCTokens-R1 | 1.3B | 0.859 | 0.250 | 0.503 | 0.592 | 0.606 | 0.604 | 0.652 | 0.308 | 0.765 | 0.337 | 0.645 | 0.324 | 0.801 | 0.353 | 0.756 |

Table 1: **Quantitative Results on Unifybench++.** TP = Text Prompt. IP = Image Prompt. Columns with * are newly extended from Unifybench. Best and second best performances are highlighted.

## 4.1 EXPERIMENT SETUP

**Experimental Details.** We first complete the first two-stage training of UniCTokens (An et al., 2025b), setting the number of learnable tokens as $K = 16$ and $M = 8$, and training for 15 epochs in each stage. The batch size for the understanding task is 4, while the batch size for the remaining tasks and text-to-image (T2I) generation is 1. In the UniCTask-GRPO stage, we set the group size $G = 9$, and the ratio of visual instruct reasoning to text-only reasoning is 1:1. All experiments are conducted on $8\times$ H100. The backbone we adopt is Show-o 512x512. More details are in the Appendix F.

**Baselines.** We compare our method with two other relative works: UniCToken (An et al., 2025b) and Yo'chameleon (Nguyen et al., 2025). For the former , we employ the identical training protocol outlined in the original paper. For the latter, we retrain the model following the specifications in its original study, using a 7B-parameter base model with 1,000 images per concept. Besides, we perform comparisons across models that specialize exclusively in either understanding or generation tasks. Furthermore, we evaluate the GPT-4o on our benchmark, regarding it as an upper bound for performance. Additional details corresponding to these baselines are provided in the Appendix F.

**Dataset.** In addition to the metrics in UnifyBench, to test the model's reasoning ability regarding personalized concepts and dence prompt personalized generation ability, we expand the Unifybench to Unifybench++. Specifically, we add Reasoning and Dense Reasoning in the Und. part. And for Gen. part, we reconstruct the Personalized-Driven-Generation into Reasoning Generation and newly establish two metrics: Dense Generation and Dense Personalized Generation. Please refers to the Appendix C to find more details about the dataset.

**Metrics.** For Reasoning and Dense Reasoning, we use BLEU score and GPT-4o to evaluate separately. As for generation tasks, GPT-4o or CLIP-T is used to determine how much the image include all the details in the explicit prompt. In particular, as for the reasoning generation tasks, exhaustive information has replaced the ambiguous information that the model originally needed to reason about.

## 4.2 MAIN RESULTS

**Results on Unifybench.** To comprehensively evaluate the personalized understanding ability, we conduct experiments on Rec., VQA, and QA tasks (Alaluf et al., 2024). As Table 1 shows, with only 1.3B parameters, UniCTokens-R1 achieves excellent performance across all these metrics with an average improvement of 12.2% compared to the previous state-of-the-art (SOTA) results. Meanwhile, we use the Pure Gen. metric for generation tasks. As shown in Table 1, UniCTokens-R1 also achieves best among unified models. Collectively, these results validate that UniCTokens-R1 provides a resource-efficient solution for personalized understanding and generation.

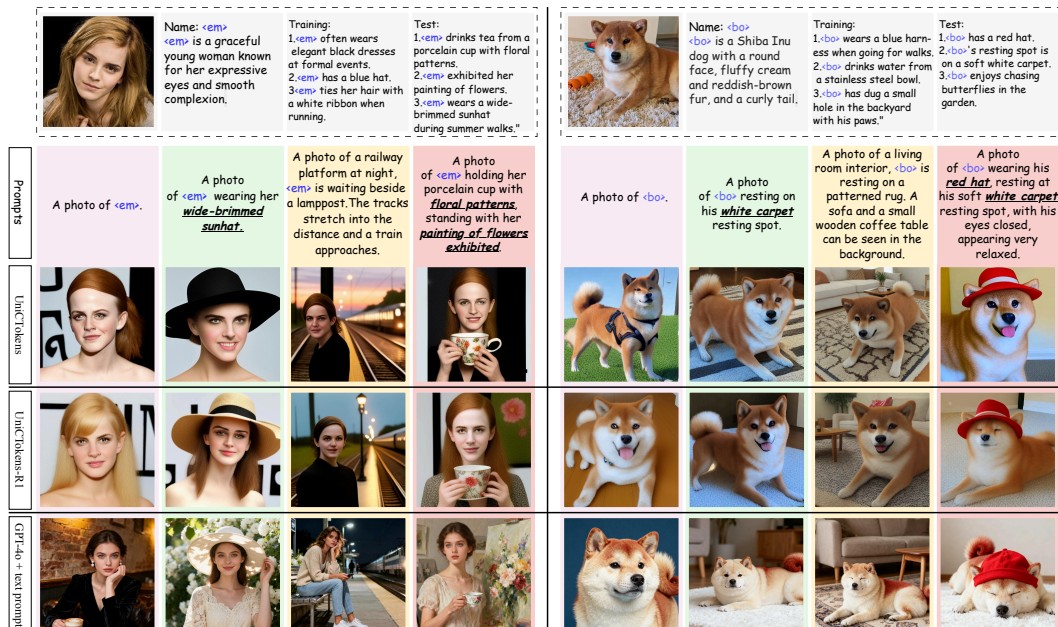

Figure 4: **Visualization Results.** We compare results generated from ours with two other methods.The underlined part is the detailed information that we expect the model to infer.

| Method | Und. | | | | | | | | Gen. | | | | | | | |
|---|---|---|---|---|---|---|---|---|---|---|---|---|---|---|---|---|
| | Rec. | Rea.* | Dense Rea.* | VQA | | QA | | Pure Gen. | | Dense Gen.* | | Rea. Gen.* | | Dense Rea. Gen.* | |
| | Weight | BLEU | GPT | BLEU | GPT | BLEU | GPT | CLIP-T | CLIP-I | GPT | CLIP-I | CLIP-T | CLIP-I | GPT | CLIP-I |
| w/o Und.Rea. | 0.808 | 0.224 | 0.427 | 0.511 | 0.519 | 0.549 | 0.621 | 0.301 | 0.762 | 0.323 | 0.637 | 0.289 | 0.765 | 0.321 | 0.727 |
| w/o VIA | 0.823 | 0.239 | 0.489 | 0.587 | 0.591 | 0.571 | 0.626 | 0.299 | 0.759 | 0.306 | 0.641 | 0.301 | 0.799 | 0.343 | 0.741 |
| w/o TIA | 0.855 | 0.222 | 0.431 | 0.572 | 0.588 | 0.597 | 0.646 | 0.305 | 0.757 | 0.319 | 0.639 | 0.291 | 0.773 | 0.329 | 0.728 |
| w/o DSOG | 0.848 | 0.243 | 0.498 | 0.590 | 0.583 | 0.607 | 0.643 | 0.305 | 0.766 | 0.331 | 0.632 | 0.307 | 0.796 | 0.338 | 0.739 |
| UniCTokens-R1 | 0.859 | 0.250 | 0.503 | 0.592 | 0.606 | 0.604 | 0.652 | 0.308 | 0.765 | 0.337 | 0.645 | 0.324 | 0.801 | 0.353 | 0.756 |

Table 2: **Ablation study on the effectiveness of the two reasoning part and DSOG method.**

**Results on Unifybench++** In Unifybench++, our main focus is on the model's performance in scenarios where it needs to reason about information related to personalized concepts and where the prompt is dense. As shown in Table 1, our method has achieved state-of-the-art results across all metrics. For the understanding tasks, especially in the Dense Rea., we have made significant progress, which strongly demonstrates that the model has a deeper understanding of concept information. For the generation tasks, compared with UniCTokens, we have achieved improvements of 14.1%, 10.0%, and 8.8% in the above-mentioned three metrics respectively. As Figure 4 shows, from left to right are the generation results of ordinary Gen., Rea. Gen., Dense Gen., and Dense Rea Gen. scenerios. Evidently, compared with the UniCTokens method, our method has achieved a certain improvement in the aesthetic level of the images. Secondly, for prompts that require reasoning, UniCTokens-R1 can correctly depict the details that need to be inferred, while the UniCTokens method cannot. In terms of the Dense generation indicators, the UniCTokens method may miss information or misplace elements in the spatial layout due to the high information density of the text. However, after training with UniCTask-GRPO, these core issues that affect image quality have been well addressed.This reflects the enhanced fine-grained performance of our method in image generation and also validates the successful mutual promotion between understanding and generation.

### 4.3 ABLATION STUDY

**Effectiveness of incorporating two-step personalized reasoning.** First, we evaluated the effectiveness of the two-step personalized reasoning. We constructed the following baselines: (1) Remove

the entire content of Personalized understanding reasoning and only optimize Personalized generation reasoning. (2) Remove the visual instruct reasoning part in personalized understanding reasoning. (3) Remove the text-only instruct reasoning part in personalized understanding reasoning.

As shown in the Table 2, when the visual instruct reasoning is removed, MMU metrics such as Rec. significantly decrease. Due to the interaction between understanding and generation, non-reasoning generation tasks also experience a serious drop. When the text-only instruct reasoning is removed, it causes severe damage to reasoning-related metrics (such as Rea. and Rea.Gen.), which proves that the model's reasoning ability is closely related to this part. If all of the personalized understanding reasoning part is removed, most metrics drop significantly. However, since the reward focuses on scoring the images, the Pure Gen. and Dense Gen. metrics do not decrease significantly compared to UniCTokens-R1. Overall, the absence of either the understanding or the generation part leads to a decline in training effectiveness, which effectively proves that the two tasks can promote each other.

**Effectiveness of DSOG.**  In addition, we also evaluated the effect of DSOG. As shown in Table 2, we can observe that under the same settings, using the DSOG method can accelerate the convergence of training, thus achieving improvements in almost all indicators at a small time cost. This fully demonstrates the effectiveness of the DSOG method.

| Method | Und. | | | | | | | | | | Gen. | | | | | | | |
|---|---|---|---|---|---|---|---|---|---|---|---|---|---|---|---|---|---|---|
| | Rec. | Rea.* | Dense Rea.* | VQA | | QA | | Pure Gen. | | Dense Gen.* | | Rea. Gen.* | | Dense Rea. Gen.* | |
| | Weight | BLEU | GPT | BLEU | GPT | BLEU | GPT | CLIP-T | CLIP-I | GPT | CLIP-I | CLIP-T | CLIP-I | GPT | CLIP-I |
| joint | 0.712 | 0.224 | 0.387 | 0.489 | 0.482 | 0.529 | 0.577 | 0.264 | 0.680 | 0.261 | 0.607 | 0.266 | 0.724 | 0.288 | 0.653 |
| joint+UniCTask-GRPO | 0.789 | 0.231 | 0.468 | 0.548 | 0.557 | 0.581 | 0.629 | 0.287 | 0.695 | 0.291 | 0.616 | 0.303 | 0.757 | 0.322 | 0.689 |
| UniCTokens | 0.792 | 0.238 | 0.385 | 0.505 | 0.521 | 0.546 | 0.601 | 0.280 | 0.750 | 0.298 | 0.639 | 0.282 | 0.762 | 0.317 | 0.712 |
| UniCTokens-R1 | 0.859 | 0.250 | 0.503 | 0.592 | 0.606 | 0.604 | 0.652 | 0.308 | 0.765 | 0.337 | 0.645 | 0.324 | 0.801 | 0.353 | 0.756 |

Table 3: **Ablation study on different initial methods.**

**Effectiveness of different initial methods.**  During our research, we found that the UniCTask-GRPO method can reduce the dependence on complex cold-start to a certain extent. Specifically, we adopted the same token design method as UniCTokens, but changed the complex multi-stage training to joint training, and then continued to train it with the UniCTask-GRPO method. The results are shown in the table. It can be easily observed that after training with UniCTask-GRPO, all indicators have improved. In the understanding tasks, there are significant improvements in the Dense Rea indicator (17.9% increase) and the VQA indicator (an average increase of 13.8%). In the generation tasks, various indicators have also improved to different degrees (an average increase of 7.5%), and scores such as CLIP-T in Pure Gen. even exceed those of UniCTokens. The above results indicate that the UniCTask-GRPO method does not rely on elaborate initialization and can be used as a universal post-training means for personalization.

## 5 CONCLUSION

In this paper, we propose the UniCTokens-R1 method, a model that achieves the integrated optimization of personalized understanding and personalized generation at the task level. Through the end-to-end reinforcement learning framework UniCTask-GRPO, we enable the understanding and generation tasks to explicitly promote each other, enhancing the model's personalized reasoning ability. This allows the model to not only further improve its basic understanding and generation capabilities but also realize the screening and application of personalized information. As a result, compared with the baseline models of UniCTokens and Show-o, the model has significantly improved in reasoning-related understanding and generation tasks as well as intensive generation tasks. To accelerate the training, we also design the DSOG method to effectively increase the group size, enabling the model to converge more quickly. In summary, this work represents a novel attempt to utilize reinforcement learning for the integrated optimization of understanding and generation in personalized tasks, providing a new paradigm for subsequent work in the field of personalization.

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

## A  THE USE OF LLMS

We leveraged large language models (LLMs) to obtain technical guidance throughout coding and debugging, and subsequently used them to edit the collaboratively drafted manuscript, improving its language and presentation.

## B  OTHER RELATED WORK

**Reinforcement learning in Large Model.**  Within the LLM domain, studies such as Chu et al. (2025) have made significant contributions to addressing key challenges including long-chain reasoning, output coherence, training efficiency, and training costs. Following the proposal of the GRPO strategy and research on the role of rule-based rewards in DeepSeek-R1 (DeepSeek-AI et al., 2025), a surge of work on MLLMs has emerged(Huang et al., 2025; Pan et al., 2025b; Zhou et al., 2025), spanning applications such as semantic segmentation(Liu et al., 2025a), object recognition(Liu et al., 2025d), video analysis(Zhang et al., 2025a; Sun et al., 2025), code generation(Austin et al., 2021; Jain et al., 2024) ,and mathematical problem-solving(Hendrycks et al., 2021; Zhang et al., 2024a;b). Ample evidence indicates that RL not only enhances in-domain reasoning capabilities but also outperforms supervised fine-tuning (SFT) in out-of-distribution (OOD) scenarios.

**Personalized model.**  The core goal of model personalization is to precisely integrate information related to specific concepts into the model output by means of diverse techniques. This technology has been explored in depth across various types of models.

In the field of generative models, recent ideas revolve around recontextualization under text conditions. For instance, DreamBooth (Ruiz et al., 2023) focuses on ensuring the authenticity of the subject in the generated results, and Textual Inversion (Gal et al., 2022) optimizes special tokens using the soft-prompt fine-tuning technique. When it comes to large language models, personalized LLM endows the model with user-aware capabilities through a dual-tower structure (Pi et al., 2024). In the field of multi-modal large language models (MLLMs), Hao et al. (2025); Alaluf et al. (2024); An et al. (2025a) enhance the personalization and relevance of the output by organically combining user information with input image content via fine-tuning or Retrieval-Augmented Generation (RAG) technology.In Unified MLLMs, Chameleon (Team, 2025) injects information using "text embedding optimization + Transformer fine-tuning" to inject information, Yo'Chameleon!(Nguyen et al., 2025) uses soft prompts, and UniCTokens (An et al., 2025a) was the first to focus on the mutual promotion between understanding and generation, significantly reducing training samples and expanding the scope of personalization tasks.

Despite the progress in customization within the Unified MLLMs field, existing methods suffer from issues such as cumbersome and time-consuming training, and poor performance in complex semantic generation tasks, which urgently need to be addressed.

**Evolution of unifying understanding and generation.**  In the academic realm, numerous studies have been conducted to enable a single model to handle both understanding and generation tasks concurrently. For example, Ge et al. (2023); Dong et al. (2024); Ge et al. (2025) aggregate language-conditioned information to drive the generation. Janus (Wu et al., 2024a) models different modalities using distinct tokenizers. Show-o (Xie et al., 2025a) and Transfusion (Zhou et al., 2024) combine autoregressive and diffusion methods to process text and images. Emu3 (Wang et al., 2024) and Selftok (Wang et al., 2025) unify multimodal data into discrete tokens and then conduct joint training.

In personalization scenarios, the cross-modal capabilities of unified models inspire our expectation to achieve complementary advantages and mutual promotion between understanding and generation. Initially, the serial combination of understanding and generation models (Luo et al., 2024; Dunlap et al., 2023; Liu et al., 2025b) posed challenges for end-to-end joint optimization. Subsequently, joint training with shared representations was attempted (Fang et al., 2023; Tong et al., 2024; Fan et al., 2024), but it merely achieved co-optimization. In this paper, we leverage reinforcement learning to amplify the mutual promotion between generation and understanding, enhance the model's reasoning ability regarding conceptual information, and improve the performance in complex semantic and reasoning-required generation tasks.

## C DETAILS OF DATASET

Building upon Unifybench (An et al., 2025a), we expanded the metrics of the original dataset and constructed Unifybench++. As shown in the Figure 5, for each concept, we created several pieces of extra_info, with each piece of extra_info corresponding to a reasoning prompt. The italicized parts are the extra_info and reasoning prompts that the model will encounter during the training process of UniCTokens-R1, while the non-italicized parts are used for Rea. and Rea.Gen. tests. In addition, we also constructed the Dense. and Dense.Rea. sections to examine the model's image-generation and reasoning capabilities in complex and dense scenarios, which correspond to the Dense Rea. metric of the understanding tasks and the Dense and Dense Gen. metrics of the generation tasks in Unifybench++.

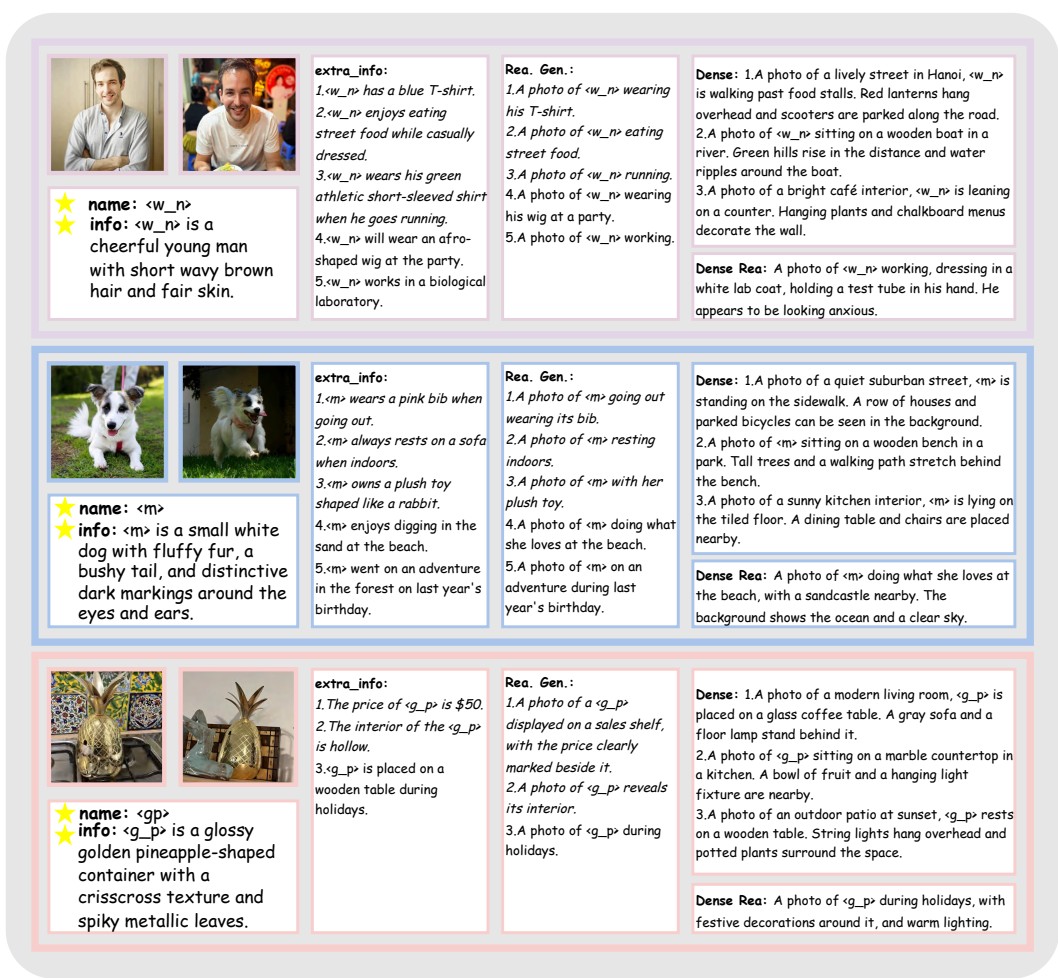

Figure 5: **UnifyBench++ Dataset.** Information of some concepts in our constructed dataset. The *italicized parts* in **extra_info** and **Rea. Gen.** are for UniCTokens-R1 training.

## D MORE DETAILS OF PERSONALIZED UNDERSTANDING REASONING

Before training UniCTokens-R1, we first use the training method of UniCTokens to preliminarily inject information related to personalized concepts. As shown in the Figure 6, after the first two-stage training, the core information about the concept will be injected into the ⟨sks⟩ token, and the semantic and image-level information of the personalized concept will be injected into ⟨token_s⟩ (16 tokens) and ⟨token_g⟩ (8 tokens) respectively.

Figure 6: **Illustration of Personalized Understanding Reasoning.** We use the tokens trained in UniCTokens method to transmit the basic information of personalized concepts.

During the Personalized understanding reasoning stage, whether it is visual instruct reasoning or text-only reasoning, we will replace $\langle sks \rangle$ with $\langle token\_s \rangle \langle token\_g \rangle \langle sks \rangle$ to convey complete concept information to the model. For visual instruct reasoning, we input the query with $\langle sks \rangle$ replaced and the image into the model to obtain the inference result $IR$. For text-only reasoning, we take the original reasoning prompt $IP$ and extra_info as inputs. The system prompt is to let the model select the item in extra_info that best matches $IP$, and finally obtain the $IR$.

## E  DETAILS ABOUT REWARDS

Our final reward is:
$$Reward = w_1 TIER + w_2 BER + w_3 DER + w_4 FER \tag{12}$$
where
$$(w_1, w_2, w_3, w_4) = \begin{cases} (0.4, 0.3, 0.3, 0) & \text{for animals and objects} \\ (0.4, 0.2, 0.2, 0.2) & \text{for humans} \end{cases} \tag{13}$$

The following is a detailed introduction to the scoring calculation methods of the four reward experts.
**Text Inference Evaluation Reward (TIER)** With ERNIE 3.0 (Sun et al., 2021), we first compute the embeddings of the ULM output $IR$ and all the concept's extra information $E_i$. Then we apply cosine similarity to get the score between $IR$ and all the extra_info $(q_1, q_2, ...q_l)$. Afterwards, we calculate the Euclidean distance $d$ between this tuple and the ground truth tuple where the ground truth tuple defines as:
$$GT = (\sigma_1, \sigma_2, ..., \sigma_l) \tag{14}$$
$$\sigma_i = \begin{cases} 0 & E_i \text{ is a wrong answer} \\ 1 & E_i \text{ is the correct answer} \end{cases} \tag{15}$$
$$\tag{16}$$
Finally, the score is calculated as:
$$TIER = \frac{1}{2}(2 - d) \quad (apparently \ 0 \leq d \leq 2) \tag{17}$$

**BLIP Evaluation Reward (BER)** With the use of the BLIP-2 model (Li et al., 2023), we first calculate the cross-modal similarity between the generated image and the text prompt:
$$s = cosine < Prompt, Image > \tag{18}$$
Then the score is calculated as:
$$BER = 1 - s \tag{19}$$

**DINOv2 Evaluation Reward (DER)** With the use of DINOv2(Oquab et al., 2023), we calculate the final score through similar methods:
$$DER = 1 - cosine < Generated \ Image, Reference \ Image > \tag{20}$$

**Facenet Evaluation Reward (FER)** For face recognition, based on MCTNN (Zhang et al., 2016), we first extract the key features of the face and transform them to the standard position. Subsequently, we calculate the embedding and obtain the final reward through cosine similarity:
$$FER = 1 - cosine < MCTNN(Generated \ Image), MCTNN(Reference \ Image) > \tag{21}$$

# F MORE DETAILS OF EXPERIMENT

**Hyperparameters.** We provide the detailed training hyperparameters in Table 4

| Name | Value | Name | Value |
|---|---|---|---|
| Group Size $G$ | 9 | Learning rate $lr$ | 1e-6 |
| $\beta$ | 0.01 | Classifier-Free Guidance Scale | 5 |
| Batchsize | 1 | Max Gradient Norm | 1.0 |
| Training Steps | 100 | Image Resolution $h \times w$ | $512 \times 512$ |
| $\alpha_{Und.}$ | 0.4 | $\alpha_{Gen.}$ | 0.6 |
| $TPR$ | 0.25 | $\mu$ | 0.12 |
| $\eta$ | 0.8 | $\varepsilon_0$ | 2 |

Table 4: UniCTokens-R1 training hyperparameters.

**Baselines.** In this part, we will introduce the baselines that are compared with our method in the experiment part.

- **Yo'LLaVA** (Nguyen et al., 2024): an early work which try to deal personalization tasks via VLM model. Following the original paper, we construct the datasets and train Yo'LLaVA with different models(Phi-1.5 (Abdin et al., 2024), 1.3B and Vicuna (Chiang et al., 2023), 13B) to make a fair comparison.

- **MC-LLaVA** (An et al., 2025a): Focused on enhancing multi-concept personalization, this model makes use of paired textual and visual prompts, standing out as a reliable baseline for personalized understanding-related tasks.

- **RAP-MLLM** (Hao et al., 2025): By leveraging the RAP-LLaVA model and following the framework of the RAP-MLLM approach, we construct a dedicated personalized database for every concept. To develop the ability to grasp personalized information, RAP-MLLM was subjected to post-training on a 260K-sized dataset.

- **Text Inversion** (Gal et al., 2022): As a specialized technique, text inversion transforms text-based prompts into corresponding visual forms, which in turn supports the generation of images from descriptive language.

- **Dreambooth** (Ruiz et al., 2023): By enabling users to fine-tune generative models with a limited image count, DreamBooth enhances the models' personalization capabilities—ultimately yielding outputs that are both highly specific and attuned to context. To ensure fair comparisons, we use different volumes of training data (10 and 3,000) to assess DreamBooth more effectively.

**Metrics.** For the metrics in Unifybench, including personalized recognition, VQA, QA, and Pure Gen. (i.e., the knowledge-driven generation metric in UniCTokens), we adopt the same calculation methods as those in UniCTokens.

For the understanding metrics in Unifybench++, the calculation method of Rea. is to compute the BLEU score between the model output and the correct extra_info. Dense Rea. is scored using GPT-4o. Regarding the generation tasks, first, we calculate the similarity between the generated image and the reference image using CLIP-I (image-to-image similarity). For non-Dense metrics (Pure Gen. and Rea. Gen.), we calculate the CLIP-T (image-to-text similarity) score between the generated image and the prompt. Here, for Rea. Gen., the prompt needs to replace the inferential part with specific details, and the same applies to the Dense Rea. Gen. metric. For Dense metrics, we segment the prompt using punctuation marks, and GPT-4o evaluates whether each segment of text is correctly represented in the image, giving a score from 0 to 1. Finally, the average score is taken.

**Results.** To ensure a fair comparison, we also assessed our method using benchmarks specifically designed for pure personalized understanding and generation. The evaluation results on the Yo'LLaVA (Nguyen et al., 2024) and MC-LLaVA Datasets (An et al., 2025a) are shown in Table 5 (left). Since our training process is not correlated with multi-concept scenarios, our tests were confined to the single-concept segment in the MC-LLaVA dataset.

| Type | Method | Yo'LLaVA | | | MC-LLaVA | | |
|---|---|---|---|---|---|---|---|
| | | Rec | VQA | QA | Rec | VQA | QA |
| | | Weight | Acc | Acc | Weight | BLEU | Acc |
| Und. Only | LLaVA+TP | 0.807 | 0.921 | 0.815 | 0.608 | 0.412 | 0.609 |
| | Yo'LLaVA | 0.932 | 0.918 | 0.897 | 0.829 | 0.657 | 0.691 |
| | MC-LLaVA | 0.953 | 0.935 | 0.924 | 0.901 | 0.692 | 0.736 |
| | Qwen2.5-VL+TP | 0.685 | 0.861 | 0.697 | 0.634 | 0.436 | 0.549 |
| | Yo'LLaVA(Phi-1.5) | 0.805 | 0.627 | 0.713 | 0.726 | 0.525 | 0.591 |
| Unified Model | Chameleon+TP | 0.715 | 0.537 | 0.702 | 0.649 | 0.408 | 0.675 |
| | Yo'Chameleon | 0.832 | 0.591 | 0.734 | 0.753 | 0.610 | 0.658 |
| | Show-o+TP | 0.704 | 0.526 | 0.605 | 0.589 | 0.574 | 0.482 |
| | UniCTokens | 0.865 | 0.602 | 0.725 | 0.742 | 0.617 | 0.692 |
| | UniCTokens-R1 | 0.926 | 0.663 | 0.737 | 0.773 | 0.671 | 0.704 |

| Type | Method | DreamBench | | Yo'LLaVA |
|---|---|---|---|---|
| | | CLIP - I | CLIP - T | CLIP - I |
| Gen. Only | Real Images | 0.873 | - | 0.864 |
| | DreamBooth | 0.692 | 0.297 | 0.645 |
| | DreamBooth | 0.815 | 0.293 | 0.788 |
| | Text inversion | 0.675 | 0.284 | 0.632 |
| Unified Model | Chameleon+TP | 0.612 | 0.168 | 0.579 |
| | Chameleon+IP | 0.594 | 0.171 | 0.499 |
| | Show-o+TP | 0.678 | 0.235 | 0.652 |
| | Yo'Chameleon | 0.807 | 0.212 | 0.771 |
| | UniCTokens | 0.788 | 0.299 | 0.806 |
| | UniCTokens-R1 | 0.808 | 0.323 | 0.822 |

Table 5: **Performance on Personalized Understanding and Generation Benchmarks.** TP = Text Prompt. IP = Image Prompt.

Our approach outperforms all the current leading unified personalized model, Significantly, our proposed UniCTokens-R1 surpasses UniCTokens (Previous state-of-the-art (SOTA) methods) by an average of 5.3% across all understanding tasks.

Also, we conduct a comprehensive evaluation of the personalized generation capabilities on the Dreambench (Ruiz et al., 2023) and Yo'LLaVA Dataset (Nguyen et al., 2024). Our methodology outshines UniCTokens in all the metrics. Significantly, with the same training data, we also outperform the prominent generative baseline, Dreambooth. This indicates that our approach is capable of generating high-quality and lifelike images incorporating user-supplied concepts.

# G  MORE QUALITATIVE RESULTS

In this section, we additionally showcase the generation results for two concepts. The highlighted parts are the elements that require the model to perform reasoning. As depicted in Figure 7 and Figure 8, our method enables the model to successfully infer detailed information related to the concepts. Moreover, it achieves a high degree of alignment in generating images from dense text.

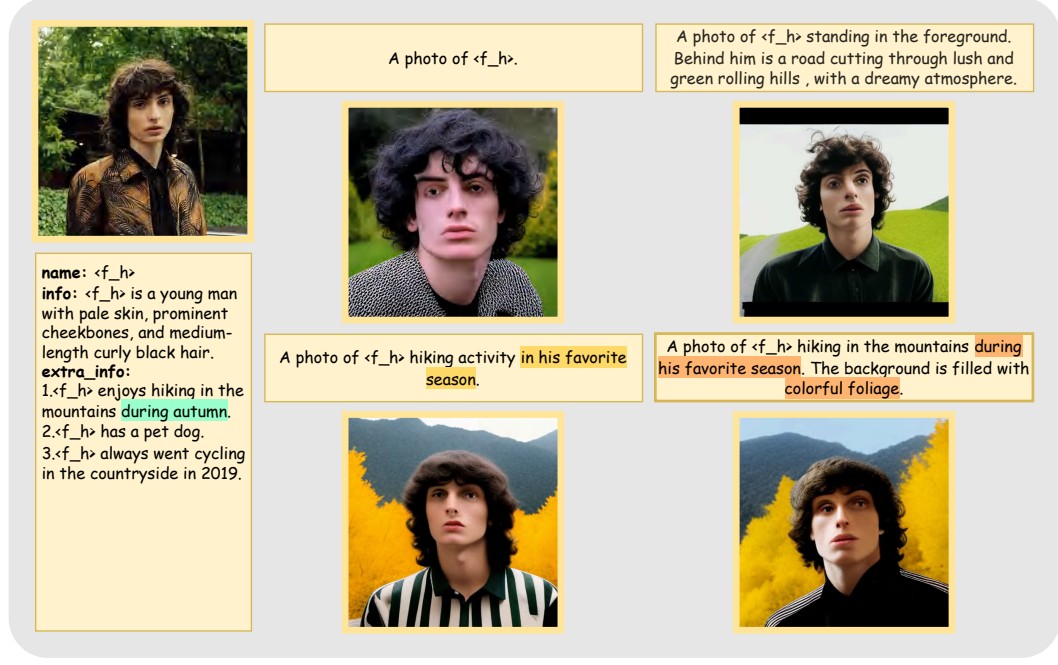

Figure 7: **Visualization Result of ⟨f_h⟩**

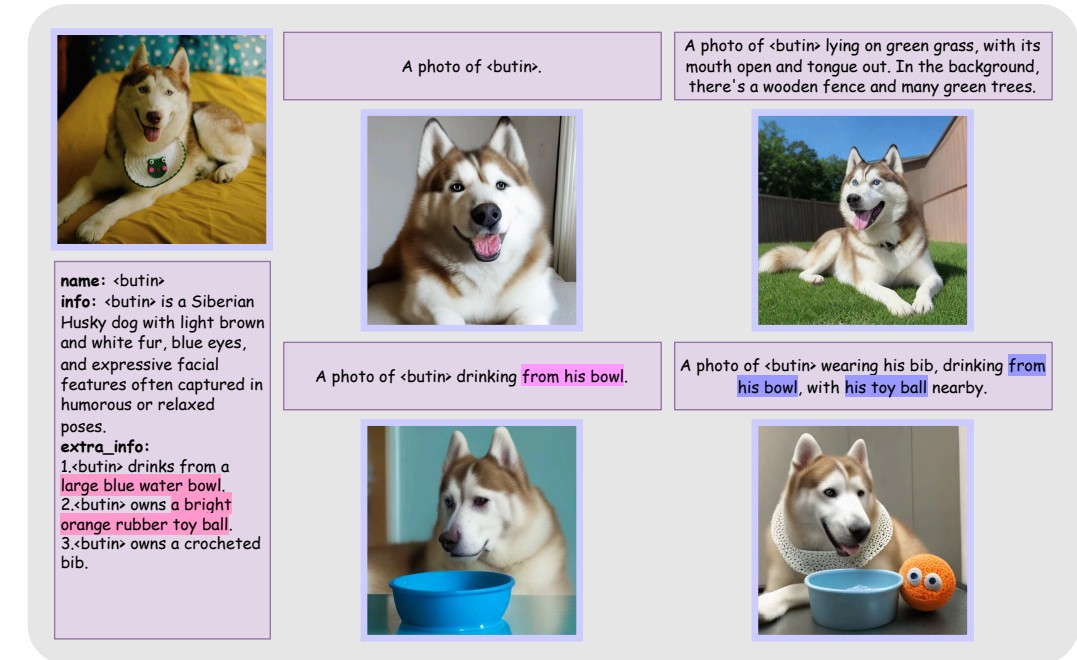

Figure 8: **Visualization Result of ⟨butin⟩**

## H  MORE DETAILS ABOUT DR.GRPO

Compared with the GRPO method, the optimization objective function turns into:

$$\mathcal{J}_{\text{Dr.GRPO}}(\theta) = \mathbb{E}_{(q,a)\sim\mathcal{D},\{o_i\}_{i=1}^{G}\sim\pi_{\theta_{\text{old}}}(\cdot|q)}$$

$$\left[ \frac{1}{G} \sum_{i=1}^{G} \sum_{j=1}^{|o_i|} \left( \min\left( D_{i,j}(\theta)\hat{A}_i, \text{clip}\left(D_{i,j}(\theta), 1-\varepsilon, 1+\varepsilon\right)\hat{A}_i \right) - \beta D_{\text{KL}}\left(\pi_\theta \parallel \pi_{\text{ref}}\right) \right) \right]$$

The $D_{i,j}(\theta)$ is the ratio between the probabilities of $\pi_\theta$ and $\pi_{\theta_{\text{old}}}$:

$$D_{i,j}(\theta) = \frac{\pi_\theta(o_{i,j} \mid q, o_{i,<j})}{\pi_{\theta_{\text{old}}}(o_{i,j} \mid q, o_{i,<j})}$$

Besides, different from GRPO, the new optimization objective function removes the normalization term $\|o_i\|$. In contrast, the advantage is calculated without the the standard deviation of the group $\text{std}\left(\{R_i\}_{i=1}^{G}\right)$:

$$\hat{A}_i = R_i - \text{mean}\left(\{R_i\}_{i=1}^{G}\right)$$

Through these modifications, Dr.GRPO has demonstrated that this approach can eliminate the impact of differences in the difficulty of problems within each batch of inputs. Additionally, it can effectively prevent the model from favoring longer outputs, which is in line with our objectives.

