# OpenReview forum: "UniCTokens-R1: Boosting Unified Personalization via Reinforcement Learning"
_ICLR.cc/2026/Conference — ICLR 2026 Conference Withdrawn Submission_

### Official Review · Reviewer_bvrx · 2025-10-31

**Soundness:** 3
**Presentation:** 3
**Contribution:** 3
**Rating:** 6
**Confidence:** 3

**Summary:**

The paper proposes UniCTokens-R1, an end-to-end reinforcement learning (RL) framework designed to improve personalized understanding and generation in unified vision-language models (ULMs). The motivation is that existing models often treat understanding and generation as separate tasks, losing opportunities for cross-task enhancement.

UniCTokens-R1 addresses this by introducing UniCTask-GRPO, an RL optimization method that jointly optimizes both personalized understanding and generation tasks in one stage. It leverages the outputs from understanding (fine-grained semantic representations) to enhance generation quality, and uses the generated outputs as feedback signals to further refine understanding.

The paper also introduces DSOG (Dynamic Scaling of Group Size) — a dynamic sampling strategy that improves convergence efficiency by filtering out low-quality generated samples during training.

For evaluation, the authors extend the UnifyBench dataset to UnifyBench++, which includes denser textual descriptions and additional user metadata to better represent real-world personalization scenarios. Experiments show that UniCTokens-R1 achieves state-of-the-art (SOTA) results across all personalization tasks on UnifyBench++, outperforming baselines like UniCTokens and Yo’Chameleon, despite using only a 1.3B-parameter model.

**Strengths:**

UniCTokens-R1 innovatively integrates personalized understanding and generation within a single RL framework (UniCTask-GRPO), enabling the two processes to mutually enhance each other rather than being optimized independently. Despite being trained on a small 1.3B model, UniCTokens-R1 achieves consistent and significant improvements across all personalized understanding and generation metrics, including reasoning and dense-generation benchmarks, showing superior reasoning transfer and generalization ability.

**Weaknesses:**

1. The reward formulation in Equation (12) combines four components, TIER, BER, DER, and FER, with different weights depending on the task category (for animals and objects vs. for humans). However, the paper does not clearly explain how these specific weights were determined. It is not stated whether they were tuned empirically, derived from validation experiments, or set heuristically.
In addition, it is unclear why such distinct weight settings are used across categories and how sensitive the model performance is to these choices. It would strengthen the paper to include an ablation study or sensitivity analysis exploring alternative configurations, such as using equal weights for all components, or employing only one reward signal at a time to assess its isolated contribution. Without this justification, the reward design appears somewhat ad hoc and leaves questions about generalizability.

2. Several minor grammatical or stylistic inconsistencies could be improved for clarity and professionalism:
* Line 54: “model However” → “model. However”
* Line 192: “shows, The” → “show, the”
* Line 205: “.However” → “. However”
* Line 210: “.The” → “. The”

**Questions:**

See the weaknesses.

---

### Official Review · Reviewer_7xWD · 2025-11-08

**Soundness:** 2
**Presentation:** 2
**Contribution:** 2
**Rating:** 2
**Confidence:** 3

**Summary:**

The paper proposes UniCTask-GRPO, an end-to-end RL framework to mutually improve understanding and generation of ULMs with the goal of improving the model’s personalized reasoning abilities. Additionally, the authors introduce DSOG, a method inspired by Shrivastava et al. (2025) that increases the group size and performs an adaptive thresholding method to accelerate the training. They also extend the existing Unifybench dataset to Unifybench++ by modifying the reasoning prompt to examine the model's abilities on personalized reasoning tasks.

**Strengths:**

- The idea is interesting, and the experiment results reflect improvement upon the existing methods.

- The ablation studies (section 4.3) are valuable and enhance the quality of the work.

**Weaknesses:**

- The paper does not have a clear problem statement, and the definitions provided in section 3.1 are more of a high-level overview of the pipeline. The training strategy and DSOG method in section 3.3 are very vaguely introduced, and heavily reliant on prior work (line 270, line 276). Some suggestions to improve section 3.3:
    1) Formally state the DSOG method.
    2) Make sure all the variables used in equations are defined in advance.
    3) Clarify the contribution over Shrivastava et al. (2025) paper
    4) Provide an analysis on why DSOG accelerates training

- The paragraph of Effectiveness of different initial methods (line 461-471) showcases some of the main strengths of this framework, but is stated very briefly towards the end of the paper. To improve the presentation, I highly recommend revising the paper and elaborating on some of the points mentioned in this paragraph earlier in the methodology sections. Specifically, the authors can highlight how this method is reducing the multi-stage training of UniCTokens to a simpler and more efficient joint training strategy while gaining performance across multiple indicators.

-------

I believe the paper in its current format has a lot of room for improvement. The two major weaknesses are the novelty and the presentation. Considering the prior work of (An et al., 2025b), the technical contribution of this work is considered incremental or not clearly stated. The paper could benefit from focusing on a few key ideas (such as DSOG, or the joint training strategy), but providing a more formal and detailed analysis. As for the presentation, I tried to provide some actionable feedback above to improve the paper.

**Questions:**

- What is the core technical contribution upon UniCTokens (An et al., 2025b) paper? The authors have reported empirical improvements in section 4.2, but in terms of methodology, it’s not clear which parts of the framework belong to the prior work, and which parts are newly proposed.

- How would different weights in the reward function (equation 11) affect the outcome and performance of the framework? I’ve read the current constant values for weights in Appendix E, but there is no explanation on how these weights were chosen.  Were any hyperparameter tuning conducted to see whether these values are optimal?

---

### Official Review · Reviewer_Yx58 · 2025-11-09

**Soundness:** 3
**Presentation:** 1
**Contribution:** 2
**Rating:** 2
**Confidence:** 3

**Summary:**

This paper studies personalized understanding and image generation tasks. The main contribution of this work is that the authors propose a 2-step inference framework (step 1: understanding; step 2: image generation) that boosts image generation quality via explicit personalized understanding. The model is updated by RL though a UniCTask-GRPO objective the authors defined. A heuristic way named DSOG(Dynamic Scaling of Groupsize) is also proposed to accelerate training. The authors demonstrate the superior performance of their framework UniCTokens-R1 compared to existing works through experiments on UnifyBench++, a augmented dataset extended by the authors based on an existing dataset UnifyBench.

**Strengths:**

1, I like the ablation study the authors conducts, which provides sufficient evidence of the role of each component in understanding/generating.

2, The experiments setup and the baseline setup are stated clearly and in detail.

**Weaknesses:**

1, The presentation needs to be improved and there are some small typos that need to be fixed. For example, in line 054, a period is missed before “However”; Line 270 has a strange indentation; In line 241, $p_\theta$ should be $p_\gamma$ instead. Also, a more clear description of Figure 2 and Figure 3 would be helpful for people to understand the algorithm. Additionally, I would suggest to state the meaning of subscripts $i,j$ clearly in Section 3.2.

2, Given that this paper builds on the idea of UniCTokens, a more detailed discussion about the difference between the two methods is needed in my opinion, and a justification about why such modification makes sense would be helpful. For example: in lines 160-162, you stated "Existing works....but this inevitably leads to information loss”, can you provide more justification about why they lead to information loss?


3, The technical contribution feels limited to me, in the sense that several critical components are adopted from existing works—for example, components of the reward model and the GRPO training methods.

**Questions:**

I have a question about the motivation of doing understanding and generating simultaneously. To me the ultimate goal for this task is generating high-quality images, regardless of understanding.

---

### Note · Authors · 2025-11-20

I have read and agree with the venue's withdrawal policy on behalf of myself and my co-authors.